# Elucidating the Anti-Tumorigenic Efficacy of Oltipraz, a Dithiolethione, in Glioblastoma

**DOI:** 10.3390/cells11193057

**Published:** 2022-09-29

**Authors:** Upasana Kapoor-Narula, Nibedita Lenka

**Affiliations:** National Centre for Cell Science, S.P. Pune University Campus, Ganeshkhind, Pune 411007, India

**Keywords:** Oltipraz, glioblastoma, cancer stem cells, apoptosis, anticancer therapeutic

## Abstract

Glioblastoma multiforme (GBM), the most aggressive primary brain tumor, displays a highly infiltrative growth pattern and remains refractory to chemotherapy. Phytochemicals carrying specificity and low cytotoxicity may serve as potent and safer alternatives to conventional chemotherapy for treating GBM. We have evaluated the anticancer effects of Oltipraz (Olt), a synthetic dithiolethione found in many vegetables, including crucifers. While Olt exposure was non-toxic to the HEK-293 cell line, it impaired the cell growth in three GBM cell lines (LN18, LN229, and U-87 MG), arresting those at the G2/M phase. Olt-exposed GBM cells induced the generation of reactive oxygen species (ROS), mitochondrial depolarization, caspase 3/7-mediated apoptosis, nuclear condensation, and DNA fragmentation, and decreased glutathione, a natural ROS scavenger, as well as vimentin and β-catenin, the EMT-associated markers. Its effect on a subpopulation of GBM cells exhibiting glioblastoma stem cell (GSCs)-like characteristics revealed a reduced expression of Oct4, Sox2, CD133, CD44, and a decrease in ALDH^+^, Nestin^+^ and CD44^+^ cells. In contrast, there was an increase in the expression of GFAP and GFAP^+^ cells. The Olt also significantly suppressed the oncosphere-forming ability of cells. Its efficacy was further validated in vivo, wherein oral administration of Olt could suppress the ectopically established GBM tumor growth in SCID mice. However, there was no alteration in body weight, organ ratio, and biochemical parameters, reflecting the absence of any toxicity otherwise. Together, our findings could demonstrate the promising chemotherapeutic efficacy of Olt with potential implications in treating GBM.

## 1. Introduction

Glioblastoma, also known as Glioblastoma multiforme (GBM) (World Health Organization grade IV glioma or astrocytoma), is the most prominent primary brain cancer associated with therapeutic resistance, high recurrence, and poor patient survival. Worldwide, an estimated 251,329 people have died from primary cancerous brain and CNS tumors in 2020 [1]. Similarly, as per the International Association of Cancer Registries, the mortality rate for GBM is ~86% every year for more than 28,000 cases reported annually in India [2]. GBM patients can have extremely poor prognoses with no current treatment modalities (for example, surgery, radiation, and chemotherapy) available that may facilitate extending median survival to a greater extent. Hence, it is considered the most aggressive and lethal multiforme. The GBM malignancy also increases by the presence of a sub-population of cancer cells, the cancer stem cells (CSCs), which are bestowed with profound tumorigenic potential. In the last decade, these cells have been identified in GBM and in a variety of cancers [3,4,5]. CSCs display the characteristics of self-renewal and multipotent differentiation potential similar to that seen in adult stem cells. Moreover, they are capable of generating new tumors with similar heterogeneity as that of the original ones. These cells are majorly responsible for aggressiveness and tumor relapse and contribute to chemo- and radio-resistance. In fact, CSCs are more resistant to conventional cancer therapies as compared to the non-CSC population, and thus are involved in the relapse of tumors. Therefore, targeting CSCs in GBM can unveil an effective chemotherapeutic strategy, which may help overcome the issue pertaining to glioma recurrence.

The conventional GBM treatment involves surgical intervention followed by chemo- and radiotherapy. However, considering the aggressiveness of GBM that causes the degradation of the extracellular matrix (ECM) of brain tissues and leads to extensive metastasis, it poses a major hindrance to the surgical excision-based complete tumor removal [6]. Hence, it is the need of the hour to explore new therapeutic compounds that can precisely target cancer cells or tumors carrying lesser side effects than existing cytotoxic/cytostatic drugs [7]. Temozolomide (TMZ) is one of the preferred drugs for treating GBM. However, the development of resistance to TMZ over a period by GBM cells does restrict its long-term administration. Moreover, it has been reported that more than half of GBM patients do not respond to TMZ due to the overexpression of DNA repair enzymes such as APNG and MGMT [8]. Therefore, there is a pressing need to explore novel and potent therapeutic options that would act as an adjuvant to radio- or chemotherapy and enhance their efficacy [5,9].

The recent data suggest the use of alternative medicine by as much as 36% of cancer patients as an adjuvant to conventional therapy [10]. Therefore, exploring new candidates that can exert anticancer potential with lesser toxicity to normal cells but can contain GBM effectively becomes imperative. In this context, investigators are channelizing their search to retrieve phytochemicals from dietary and medicinal plants to use those as probable preventive and chemotherapeutic agents in treating cancer, more so because of their pro- and antioxidant activities based on the concentrations used. In normal cells, they serve as free radical scavengers when used at a lower concentration. Their antioxidant effects help maintain redox homeostasis, which is crucial for different cellular processes and disease-fighting efficacy. However, in cancer cells with very high levels of reactive oxygen species (ROS) than the normal cells, the phytochemicals at higher concentrations act as prooxidants and induce autophagy, apoptosis, and necroptosis via mTOR activation and PI3-Akt inhibition [11]. In recent years, the “integrative” oncology society has mandated advances from scientists to develop natural medicine [12]. Thus, they are considered to be a treasure house for developing novel drugs. 

Oltipraz [5-(2-pyrazinyl)-4-methyl-1, 2-dithiol-3-thione] (Olt) is a synthetic dithiolethione found among the members of the Cruciferous family. The Olt and its oxidized metabolites carrying proven strong antioxidant effects have been studied in various diseases such as obesity, type 2 diabetes, heart failure, kidney injury, and renal and liver fibrosis in preclinical and clinical settings alike [13,14]. Jiang et al. [14] have demonstrated its role as an ROS scavenger that prevents glucose-induced oxidative stress generation and apoptosis in Schwann cells via Nrf2/NQO1 signaling. The preclinical evaluation of Olt has revealed its tumor-suppressive efficacy in breast, bladder, colon, stomach, liver, lymph nodes, lung, pancreas, and skin cancers induced by different carcinogens [15]. Phase I and II trials concerning the Olt in colon, breast, liver, and lung cancer patients have also revealed its anti-tumorigenic potential together with its minimal toxicity to normal cells [15]. However, its effect on human GBM cells or CSCs is yet to be explored. 

In this study, we have used Olt to assess its efficacy in containing human GBM cell growth both in vitro and in vivo to gain new insights into the possible mechanism underlying the cytotoxicity activity of Olt in GBM cells. Here, we investigated the effects of Olt on cell viability, migration, invasion, and apoptotic-inducing properties in vitro and in a xenograft mouse model in vivo. We found that Olt leads to apoptosis in GBM cells and represses the expression of the vital genes associated with the CSC phenotype. Hence, it decreases the stemness properties of CSCs by directing them towards differentiation. Moreover, oral administration of Olt did not render systemic toxicity in mice. Thus, we propose that OLT may be a novel promising target for GBM therapy.

## 2. Material and Methods

### 2.1. Cell Culture and Treatments

Three glioblastoma cell lines (U-87 MG: ATCC Cat # HTB-14; LN-18: ATCC Cat # CRL2610; LN-229: ATCC Cat # CRL-2611) representing the grade IV GBM and HEK-293 (ATCC Cat # CRL-1573), the human embryonic kidney epithelial cells, were obtained from the cell repository of the National Centre for Cell Science (NCCS), Pune. We also considered the mesenchymal stem cells (MSCs) derived from gingival tissues, already available in the laboratory for comparative studies. MSC isolation was conducted prior to this study according to the guidelines of the Declaration of Helsinki (collected at AFMC, Pune, India with informed patient consent and Institutional Ethical Committee approval # NARI/RSP/12-13/). The HEK-293 and U-87 MG cells were maintained in culture using MEM-α with 10% FBS and 1 mM sodium pyruvate (all from Invitrogen) at 37 °C in an incubator with controlled humidity and a 5% CO_2_-95% air atmosphere. For maintaining LN-18 and LN-229 in culture, MEM-α was substituted with DMEM. Similarly, MSCs were maintained in DMEM supplemented with 10% FBS, 2 mM L-glutamine, 1× non-essential amino acid, 100 IU/mL Penicillin:Streptomycin (all from Invitrogen, Carlsbad, CA, USA), and 100 µM β-ME (Sigma–Aldrich, St. Louis, MO, USA). The propagation of cells was carried out at regular intervals after attaining 80–90% confluence. Various concentrations (20–60 µM) of Olt (Sigma–Aldrich) were used to assess the dose–response on cell growth parameters. 

### 2.2. Cell Viability Assay

Viability is a measure of the metabolic state of a cell population that indicates the growth potential. To examine the effects of Olt on control non-tumorigenic cells (HEK-293; MSC) and various GBM cells (LN-229, LN-18, U-87 MG) in culture, cell viability was measured by MTT assay, following the conventional protocol. HEK-293 cells with epithelial morphology similar to the stated GBM cells were used as control cells for comparison. Similarly, MSCs were considered as control cells since they give rise to myofibroblasts in the tumor microenvironment. Briefly, ~10,000 cells/well were seeded in a 96-well plate either with or without specified concentrations of Olt. After 24 and 48 h of incubation, we assessed the cell viability by incubating the cells with 10 µL of MTT (5 mg/mL) (Sigma–Aldrich) for 4 h at 37 °C, followed by the addition of 100 µL DMSO and further incubation for 1 h to dissolve the purple formazan crystals. The absorbance was taken at 570 nm using Spectramax 384 microplate reader (Molecular Devices, San Jose, CA, USA). The value of untreated cells (Ctrl) in each cell line was considered as 100%, and the comparative percent viability with Olt treatment was calculated accordingly. 

### 2.3. Colony Formation and Cell Migration Assay

5000 cells/mL were taken in the treatment and control groups and incubated for 14 days with intermittent medium change. Colonies formed in all groups were stained with crystal violet (0.05%) and imaged. The migration assay was performed using the Transwell Boyden chamber (Corning) following the protocol [16]. Briefly, U-87 MG cells (1 × 10^5^) pretreated with the Olt were placed in the upper chamber, while the lower chamber contained 10% FBS as a chemo-attractant. After 12 h of incubation at 37 °C in a CO_2_ incubator, the cells that had migrated to the lower surface of the Transwell membrane were fixed with 4% paraformaldehyde (PFA) for 10 min. Further, the staining was carried out with 5% Crystal Violet (prepared in 25% methanol) for 10 min, followed by washing to remove the unbound stain. Migrated cells were photographed under an inverted microscope (TE2000-U, Nikon, Kawasaki, Japan).

### 2.4. Flow Cytometry Analysis

#### 2.4.1. Cell Cycle Analysis

Cellular DNA content was studied to analyze the cell cycle progression by flow cytometry using propidium iodide (PI) as the DNA dye [17]. Briefly, the cells (0.5–1 × 10^6^) were fixed in ethanol, washed with PBS, and incubated for 30 min in the dark at room temperature with the staining solution containing RNase A (100 μg/mL) and PI (25 μg/mL). Flow cytometry (FACS Calibur, Beckton Dickinson, San Jose, CA, USA) was carried out to analyze the cell cycle, wherein PI fluorescence was measured through an FL-2 filter (585 nm).

#### 2.4.2. Measurement of ROS

We have used the cell-permeable fluorescent dyes DCFDA (Thermo Fisher Scientific, Carlsbad, CA, USA) and DHE (Sigma–Aldrich, St. Louis, MO, USA), respectively, to quantify superoxide (O_2_^−^) and hydrogen peroxide (H_2_O_2_), the known subtypes of ROS [16]. Briefly, cells treated with phytochemicals for 48 h were trypsinized, washed, suspended in DHE (5 μM) or carboxy-DCFDA (5 μM), and incubated for 30 min in the dark at 37 °C. Subsequent to PBS wash, cells were resuspended in PBS, passed through a cell strainer, and quantified by flow cytometry (FACS Calibur, Beckton Dickinson, San Jose, CA, USA) using the CELL Quest PRO^TM^ (v 5.2.1, Becton Dickinson, San Jose, CA, USA) software. Around 10,000 cells were examined for each sample. Unstained cells served as a negative control for background correction, and the untreated fluorophore-loaded cells could serve as a control (Ctrl).

#### 2.4.3. Changes in Mitochondrial Membrane Potential (MMP: ∆ψm)

One of the major changes observed in apoptotic cells is the depolarization of the inner membrane of mitochondria. Hence, MMP was measured using Rhodamine 123 (Rh123) (Sigma–Aldrich), a fluorescent dye that binds to the inner mitochondrial membrane and is released upon membrane depolarization [18], to detect the Olt-induced apoptosis in U-87 MG cells. Briefly, U-87 MG cells (Ctrl and treated) were incubated in the dark with 5 µg/mL of Rh123 for 30 min at room temperature. The signal was assessed by flow cytometry using FL-1 channel (590 nm band-pass filter) for detection. 

#### 2.4.4. GSH Estimation

Cell tracker green (CMFDA, Thermo Fischer Scientific, Carlsbad, CA, USA) was used to measure the total GSH, the major cellular thiol in cells, as described [19]. Briefly, the cells were incubated with the CMFDA (5 µg/mL) for 30 min in the dark at 37 °C. The treated cells were then acquired on the flow cytometer, and analysis was performed using the CELL Quest PRO^TM^ software.

#### 2.4.5. Assessment of Apoptosis Induction

(a) Annexin-PI assay: U-87 MG cells (ctrl and treated) were harvested after 48 h, and apoptosis was detected by staining with Annexin V-APC and PI [20]. Briefly, cells were trypsinized, washed with PBS, and then incubated with AnnexinV-APC (bd Biosciences) in the dark for 15 min. PI was added just before acquiring samples to differentiate between necrotic and live cells by flow cytometry. The data were analyzed using CELL Quest PRO^TM^ software to determine the percentage of apoptotic cells present. 

(b) Caspase 3/7 expression: Caspase 3/7 expression was assessed by flow cytometry using a commercially available CellEvent Caspase-3/7 Green Detection kit (Invitrogen) following the manufacturer’s instructions. Briefly, cells were harvested, added to 500 nM solution of Caspase-3/7 Green Detection Reagent, and incubated at 37 °C for 25 min. Finally, 1 μM of SYTOX™ AADvanced™ stain solution was added to the reaction mix, incubated for 5 min, and quantified by flow cytometry.

(c) Hoechst 33242 staining and assessment of nuclear morphology: Hoechst 33342 (Thermo Fisher) that binds to DNA was used to identify any changes in the nuclear morphology [21]. Hoechst (5 mg/mL) was added to cells and incubated at 37 °C for 15 min. Cells were visualized under a Zeiss LSM510 META microscope (Carl Zeiss, Germany), and the image acquisition and analysis were carried out using LSM 5 Image Browser software (v AIM 4.2.0.121, Carl Zeiss, Oberkochen, Germany).

### 2.5. Analysis of Stem Cell Markers 

#### 2.5.1. ALDH Assay

ALDH^+^ cells were quantified using the ALDEFLUOR Kit from Stemcell Technologies, Vancouver, as per the manufacturer’s protocol. Around 5 × 10^6^ cells (U-87 MG) were considered for ALDH analysis and the sorting of the ALDH^+^ population using the FACSAria III instrument (Becton Dickinson, Franklin Lakes, NJ, USA). Sorted ALDH^+^ cells were exposed to Olt (40- and 60 µM) for 48 h and quantified further by flow cytometry to detect the influence of Olt on the ALDH^+^ population. The results were analyzed using FACS-Diva software (BD FACS-Diva v 6.1.3, Becton Dickinson, San Jose, CA, USA).

#### 2.5.2. CD44 Estimation

U-87 MG cells were sorted based on the expression of CD44, the CSC marker, on their surface [22]. Cells were trypsinized and incubated with the primary antibody (Abcam; 1:1000 dilution) for 60 min at 4 °C, followed by thorough washing and binding with PE-conjugated secondary antibody (1:200 dilution) for another 30 min at 4 °C. Following thorough washing with PBS, the CD44^+^ and CD44^−^ cells were purified by FACS and seeded under maintenance conditions for 12 h for their adhesion and growth. The same were subjected to Olt (60 µM) exposure for 48 h. Subsequently, the cells were trypsinized and quantified for CD44 expression by flow cytometry. 

#### 2.5.3. Flow Cytometry Quantification of Nestin^+^ and GFAP^+^ Cells

The Olt-treated cells were subjected to flow cytometry-based quantification of Nestin^+^ and GFAP^+^ cells following the protocol described [23]. Briefly, cells were trypsinized, fixed with 4% PFA for 20 min on ice, permeabilized using 0.05% saponin, and incubated at 4 °C with the appropriate primary antibody (1:200) in 1% blocking buffer for 60 min. Following thorough washing, the cells were incubated with PE-tagged secondary antibody (1:200) in 1% blocking buffer for 30 min at 4 °C. Cells were given PBS wash and acquired using FACS Calibur. The background fluorescence was subtracted using appropriate negative controls.

### 2.6. Immunocytochemistry

U-87 MG cells (Ctrl and Olt-treated) were seeded on cover-slips coated with gelatin. After 48 h of incubation, immunocytochemical analysis was performed following the standard protocol [24]. Antibodies used were against α-tubulin (Sigma–Aldrich; 1:1000), GFAP (Sigma–Aldrich; 1:500), γH2AX (Abcam; 1:1000), CD44 (Abcam; 1:1000), CD133 (Abcam; 1:1000), Sox2 (Santacruz; 1:500), Oct-4 (Santacruz; 1:500), Nanog (Santacruz; 1:200), β-catenin (Santacruz; 1:200), Vimentin (Santacruz; 1:200), Nestin (Millipore; 1:200), and corresponding Cy3-conjugated secondary antibodies (Millipore; 1:500). Images were captured with a confocal laser scanning microscope (Zeiss LSM510, Germany) and analyzed.

### 2.7. Formation of Primary and Secondary Oncospheres

To assess the stemness and clonogenicity of plausible glioma stem cells (GSCs) present in U-87 MG cells and the influence of the Olt on them, an oncosphere formation assay was carried out using hanging drop method. Cells (control and treated) were considered for hanging drop preparation at a density of 500 cells/20 µL in the maintenance medium [24]. After 48 h of incubation, the spheres designated as primary (1°) oncospheres were collected and maintained in suspension culture for 3 days. Subsequently, they were enzymatically dispersed and subjected to secondary (2°) sphere formation by the hanging drop method. The number of (1°) and (2°) oncospheres generated were manually counted under the microscope, and the area of each was determined.

### 2.8. Ectopic GBM Model In Vivo

All animal experiments were carried out as per the institutional guidelines and following a protocol approved by the Institutional Animal Ethics Committee (IAEC), NCCS, Pune, India (Approval Code: EAF/2017/B-280 dated 13 March 2018). U-87 MG cells (0.5 × 10^6^) suspended in 100 μL sterile PBS were injected subcutaneously into the right flank of 4–5 wk old male SCID mice housed at the experimental animal facility (EAF) of NCCS. The mice with palpable tumors (~1 wk) were randomly divided into different groups with five animals each (n = 5) [Group (a): vehicle control; group (b and c): Olt (100 and 150 mg/kg body wt. respectively); and group (d): TMZ (50 mg/Kg body weight), as the positive control]. After completion of the experiment, blood was collected by retro-orbital route; the mice were sacrificed, and tumors and all vital organs were collected for hematoxylin and eosin (H&E) staining. We measured the tumor size in each using a Digital Vernier Caliper and calculated the respective tumor volume (mm^3^) using the formula A × B^2^ × 0.52 (A = length; B = width; all parameters in millimeters) [22]. Both blood and serum were used for hematological and biochemical analysis, respectively. The tissues were fixed using 4% PFA at 4 °C overnight, dehydrated with 20% sucrose, embedded in paraffin, and finally taken for sectioning. Around 5 µM slices from each were mounted on glass slides for H&E staining. 

### 2.9. Statistical Data

All data were presented as mean ± SEM (standard error of the mean) with a minimum of 3 independent experiments, as mentioned in the figure legend; statistical significance was calculated using paired/unpaired Student’s *t*-test (SigmaPlot v10.0, SysStat Software Inc., Palo Alto, CA, USA) compared to control. “*p*” values were calculated and are represented as follows: * *p* ≤ 0.05; ** *p* ≤ 0.01, *** *p* ≤ 0.001.

## 3. Results

### 3.1. Olt Alters the Morphology and Growth of GBM Cells

To determine the effect of Olt on the cell growth characteristics, the MTT assay was carried out using the conventional chemotherapeutic drugs cisplatin (Cspln) and temozolomide (TMZ) as positive controls. The chemical structure of Olt is given in Figure 1a. As seen in Figure 1b, there was a significant impairment in cell viability in the case of GBM cells following their exposure to Cspln, TMZ, or the Olt. However, none of these impaired the cell viability in the case of HEK-293 cells and MSCs, indicating a lack of any adverse effects on normal cells. Interestingly, MSCs showed significantly higher viability with 20 μM Olt exposure (129.42 ± 8.86) compared to the untreated control (Figure 1b). In contrast, the relative viability of treated GBM cells decreased significantly as the concentration of Olt increased. While ~65–75% of cells were viable with 20 μM Olt exposure among various GBM cell lines studied, the same was 53–64% and 43–58% with 40- and 60 μM Olt exposure, respectively. Strikingly, most of the cells from the GBM group got dislodged with 80 μM Olt exposure by 48 h (Figure 1c). This prompted us to consider 40 and 60 μM as the chosen doses of Olt for subsequent experiments using U-87 MG, one of the grade-IV GBM cell lines. Moreover, the inhibitory effect of Olt was maximum at 48 h. compared to 24 h (data not shown), and most cells were found necrotic by 72 h time point (Figure 1c). Hence, we chose the 48 h time point for all the subsequent experiments. 

To determine the effect of Olt on the morphology of U-87 MG cells, various concentrations (40- and 60 μM) of Olt exposure were given to cells, followed by monitoring of the morphological alterations under a microscope. As shown in Figure 1c, most of the untreated cells adhered to the tissue culture dish retaining an intact cytoskeleton. However, the cells treated with 40- and 60 μM Olt displayed a round and shrunken morphology similar to that with 30 μM Cispln. Moreover, the cells started dislodging from the dish and forming floating aggregates with 60 μM Olt after 48 h of exposure. A dramatic decrease in viable cell numbers with atypical apoptotic and necrotic morphology was detected among the remaining adherent ones. 

### 3.2. Olt Impaired Migration and Colony Formation in GBM Cells

The treatment with the Olt at both the stated concentrations decreased the number of colonies and their size compared to Ctrl (Figure 1d). This finding suggested the efficacy of Olt in rendering anti-tumorigenicity. In fact, a negative correlation was noted between the increasing concentrations of Olt with that of the ability to form a colony in the soft-agar assay. Further, considering the highly aggressive and metastatic nature of GBM, the effect of Olt on cancer cell migration was studied using a trans-well migration assay. As seen in Figure 1e, Olt could attenuate the migration of U-87 MG cells. 

### 3.3. The Mode of Action of Olt in Rendering Growth Inhibition in GBM Cells

To validate further the Olt responsive GBM cell growth impairment and the causative thereof, the cell cycle pattern was analyzed in U-87 MG cells treated with 40- and 60 μM Olt for 48 h. As seen in Figure 1f and Appendix A, the Olt exposure could arrest U-87 MG cells at the G2/M phase. While an increase in the G2/M population to 15.5 ± 2.49% could be seen with the 40 μM Olt, the same with 60 μM was 20.38 ± 3.48%. On the contrary, the Olt exposure led to a decrease in the G1 and S phases of the cell cycle. This may reflect the association between G2/M phase cell cycle arrest and reduction in GBM cell viability. 

Moreover, staining with annexin V-APC and PI followed by flow cytometry quantification revealed a significant increase in the apoptotic cell population in the Olt treated group (Figure 2a and Appendix A). While there was a significant decrease in live population (Q1: Annexin V^−^/PI^−^) in the treatment group, the opposite was true for the early (Q4: Annexin V^+^/PI^−^) and late (Q3: Annexin V^+^/PI^+^) apoptotic population. However, a significant difference was not noted with respect to the necrotic population represented in Q2 (Annexin V^−^/PI^+^). Caspase 3/7 assay further validated this, showing increased Caspase 3/7^+^ cells with Olt exposure (Figure 2b). Collectively, our experimental evidence demonstrated that Olt might induce caspase-dependent apoptosis in human GBM cells.

Further verification of apoptosis was carried out using chromatin condensation and nuclear fragmentations assays, the hallmark characteristics of apoptotic cell nuclei by staining with Hoechst 33412 and phosphorylated H2A histone family member X (γH2AX -a marker of DNA double-strand break: DSB) (Figure 2c,d). The Olt exposed cells exhibited typical apoptotic features such as condensed chromatin, nuclear fragmentation, and membrane blebbing (Figure 2c). DSB was also evident by a notable increase in γH2Ax expression following exposure to the Olt. As seen in Figure 2d, there was a significant increase in punctate structures in the Olt-treated cells’ nuclei. This was accompanied by a loss of cell integrity, as ascertained by monitoring microtubules disorganization through α-tubulin staining (Figure 2e). Olt exposure caused microtubule disruption in U-87 MG cells, as seen by compact and dense cellular microtubule bundles surrounding the nuclei compared with untreated Ctrl cells. 

### 3.4. Olt Renders ROS-Dependent Apoptosis by Reducing Intracellular GSH Content and Disrupting Mitochondrial Membrane Potential

To ascertain if Olt-induced apoptosis was ROS-dependent, the levels of oxygen free radicals were determined by staining with DCHFDA and DHE. As seen in Figure 3a,b, there was ~3–5-fold increase in ROS in cells exposed to the Olt. In fact, the increase in the percentage of ROS-positive cells indicated a significant generation of free radicals/oxidative stress with the treatment of Olt, irrespective of the concentration used. There was also a loss of MMP, another hallmark of apoptosis (Figure 3c), along with a decrease (~3 folds) in GSH level (Figure 3d), compared to that of control. Taken together, these data demonstrated that Olt-induced apoptosis in GBM cells was manifested through mitochondrial dysfunction and ROS generation.

### 3.5. Olt Attenuates the Growth of GSCs and Induces Their Differentiation In Vitro

Glioma stem cells (GSCs) show a characteristic CD44^+^ phenotype and enhanced ALDH activity. To investigate the effect of Olt on these subpopulations of brain cancer cells, U-87 MG cells were stained with either CD44 or ALDH. Subsequently, the purification of both positive (ALDH^+^: ~55–75%; CD44^+^: ~75–90%) and negative populations was carried out in each by FACS. The sorted cells were further seeded both in the presence and absence of Olt and maintained in culture to assess the characteristics in each. Interestingly, ALDH^+^ cells post-seeding with Olt exposure exhibited a significant reduction in ALDH activity showing around 47% and 45% ALDH^+^ cells with 40- and 60 µM Olt, respectively (Figure 4a). There was also overall growth retardation with Olt exposure. The same was true in the case of CD44^+^ GSCs as well, which showed a reduction of around 20% CD44^+^ cells upon Olt exposure at 60 µM concentration (Appendix A), suggesting the efficacy of the Olt in retarding CSCs growth. Intriguingly, CD44^−^ cells that were supposedly representing the non-CSCs regained CD44^+^ (43%) status post-seeding, albeit with a significant reduction in them upon Olt exposure (27%). This suggested the plausible occurrence of either dedifferentiation or the presence of primed GSCs in the CD44^−^ fraction. Together these data suggested the Olt to be effective in containing GSCs and non-GSCs alike.

One way to contain cancer would be to target the CSCs and induce them to differentiate. Hence, we analyzed the expression of CD44 in publicly available glioblastoma TCGA datasets using the ULCAN. CD44 expression was seen to be significantly enhanced in primary brain tumors as compared to normal brain tissues. In addition, patients with high CD44 expression had shorter overall survival when compared to patients with low CD44 expression (Appendix A). Accordingly, we compared the expression of various GSC and differentiation-specific markers in untreated control and the Olt-treated cells. While the number of nestin^+^ cells decreased with Olt treatment (Figure 4b), the reverse pattern was observed concerning GFAP, a differentiated astrocytic marker (Figure 4c). This was further validated by monitoring the expression of these markers by immunocytochemical analysis. We noticed a significant reduction in the expression of CD44, Sox2, Oct4, and Nanog in the Olt-exposed U-87 MG cells suggesting impairment in stemness characteristics (Figure 4d,e and Appendix A). In contrast, an increased expression of GFAP in Olt exposed cells (Figure 4f) suggested that the Olt might have directed the GSCs to undergo differentiation, thereby curbing their self-renewal potential.

The ability to form oncospheres is used to assess the self-renewal capacity of CSCs. Considering that GSCs possess the capability to generate oncospheres in non-adherent culture, we developed the oncospheres by using the hanging drop method. Olt exposure could reduce the sphere-forming ability in U-87 MG cells, as ascertained by monitoring the number of oncospheres generated and their area. Treatment with the Olt significantly compromised the sphere-forming ability, as depicted by a significant decrease in the number of the primary (1°) and secondary (2°) oncospheres (Figure 4g,h). The same were further characterized by studying the expression of GSC markers, CD133 and CD44, in them. As seen in Figure 4i, a striking decrease in expression of CD133 and CD44 was seen in the Olt-treated (1°) oncospheres as compared to Ctrl. 

### 3.6. Olt Attenuates the Epithelial to Mesenchymal Transition

Because EMT is another hallmark of cancer metastasis, we investigated the effect of Olt on EMT markers, i.e., β-catenin and vimentin, after treatment with the Olt. There was a significant reduction in the expression of β-catenin and vimentin in Olt-treated cells compared to control (Figure 5a–d). These results suggest a plausible role of Olt in modulating EMT via β-catenin and vimentin expressions. Further work in this context would indicate the underlying EMT mechanism.

### 3.7. The Effect of Olt Administration on the Ectopic GBM Mouse Model

Since our data suggested the anti-tumorigenic potential of Olt in U-87 MG cells in vitro, we were interested in assessing its efficacy in vivo as well by establishing an ectopic GBM model. The schematic for in vivo experiment is given in Figure 6a. Palpable tumors were detected in SCID mice after 1 wk of injection of U-87 MG cells in them. There was a significant reduction of more than 40% in tumor volume and weight in the Olt administration group compared to the Ctrl (Figure 6b–d). The TMZ was used as a positive control in these experiments. However, there was no change in the body weight in the Olt- and TMZ-treated groups compared to the vehicle control group (Table 1). Similarly, no significant difference was noted between the organ body weight ratio of different vital organs such as brain, liver, kidney, lungs, and spleen in treatment as well as Ctrl groups, suggesting no induction of systemic toxicity, if any, with the administration of Olt (Table 2). In contrast, there was a significant increase in spleen weight in disease control compared to healthy control (Table 2). Furthermore, concerning the biochemical (SGOT, SGPT, ALP, Albumin, Creatinine, blood urea nitrogen) and hematological parameters, no striking difference was noticed in any of the groups tested (Table 3). Similarly, the H&E staining of respective tissue sections revealed no evident histopathological abnormalities in any of the vital organs (brain, heart, liver, kidney, and lungs) tested (Figure 6e). However, extensive necrosis with occasional nuclear debris could be seen in tumor slices of Olt- and TMZ-treated groups compared to the disease control group. Together, our data suggested the efficacy of Olt in containing GBM similar to that of TMZ. 

## 4. Discussion

GBM, usually found in the cerebral hemispheres, is considered as one of the most aggressive tumors of the brain. It consists of poorly differentiated astrocytes having an extremely proliferative and invasive nature. Considering the survival time for GBM patients is very low, and that ~95% of patients die during the early months of diagnosis, identifying the prognostic factors related to GBM becomes difficult. The widely used chemotherapeutic drug, namely TMZ, has also been found to be associated with chemo-resistance against GBM treatment. This is primarily due to the domineering effect of the DNA repair enzyme O6-methylguanine-DNA methyltransferase (MGMT), which negates the TMZ-induced DNA alkylation [25]. Undoubtedly, the emerging drug resistance leads to higher mortality. Hence, the need of the hour is to develop suitable anticancer drugs for the treatment of GBM that should have fewer side effects, and that can act as adjuvant drugs to improve the efficacy of surgery and chemotherapy along with potency to avert tumor relapse [26,27].

One of the plausible strategic modalities for new drug screening would involve inhibition of proliferation, migration, and invasion of GBM cells. Accordingly, in the present work, we have evaluated the GBM tumor-specific anti-proliferative and cytotoxic activities of the dithiolethione Olt and its anti-tumorigenic efficacy both in vitro and in vivo. Several studies have shown that dithiolethione can decrease cell viability in lung and breast cancers [28,29]. In human metastatic breast cancer cells, Olt was shown to inhibit growth, induce the formation of apoptotic bodies, and increase DNA fragmentation and laddering [30]. Moreover, in various carcinogenesis models, Olt was seen rendering protection from the induction of cancers in the skin, breast, bladder, lung, colon, pancreas, stomach, and liver [15]. Our results did suggest a differential response of Olt in normal and different grades of GBM cell lines. While a significant reduction in the viability of GBM cell lines accompanied by G2/M arrest was apparent following Olt exposure (40- and 60 µM), the same in MSCs and the HEK293 cell line remained unaltered compared to the untreated control. Indeed, the higher viability of MSCs at a lower concentration (20 µM) of Olt reflected its antioxidant effect in normal cells in contrast to the prooxidant effect in GBM cells. Natural compounds such as withaferin and gastrodin have also been shown to decrease the cell viability of GBM cells by arresting cells at the G2/M phase [31,32]. Similar results were reported with dioscin, a natural compound isolated from *Polygonatum sibiricum,* which led to the cell cycle arrest of HepG2 cells at the G2/M phase [33]. Moreover, a hybrid of Olt has also been shown to cause G2/M cell cycle arrest and induce apoptosis in human leukemia HL-60 cells [17]. Hence, Olt might be exerting its tumor-specific cell viability influence via G2/M arrest. 

Since cell migration is a key process to the underlying immune response, wound healing, invasion, and metastasis, microtubules play a crucial role in mitosis and cell division, and these factors have emerged as important targets in cancer therapy [34]. In our study, we found that Olt significantly attenuated the ability of GBM cell migration. Moreover, a noticeable disarrangement of microtubules suggested that Olt decreased the metastatic potential of GBM cells. Similar results were obtained by Hirtz et al. [35], showing GPER agonist G1 promoting cell death and a decrease in cell viability in LN-229 and U-251 cells via the modulation of microtubule dynamics.

Several key parameters, such as increased ROS, decreased MMP, and glutathione level, activate intrinsic apoptosis and, thus, direct cell death [35]. GSH is the most critical thiol-containing molecule and functions against oxidative stress. Indeed, the balance between intracellular ROS and GSH levels controls cell death in cancer cells [36]. Excessive buildup of ROS leads to change in mitochondrial functions and activates a series of mitochondria-associated events corresponding to apoptosis [36]. Moreover, ROS accumulation can cause depolarization of the mitochondrial membrane and thereby promote apoptosis [27,37]. It has also been proposed that ROS generation plays an essential role in the apoptotic cell death induced by cytotoxic drugs [38]. In agreement with these concepts, our findings demonstrate that Olt induces apoptosis in GBM cells through the disturbance of MMP and enhanced ROS production with a concomitant reduction in GSH level. Moreover, the detection of punctate nuclei via γH2αX staining could also validate nuclear condensation and fragmentation, the characteristic apoptotic features, in Olt exposed cells. Similar results were obtained by Liang et al. [32], who showed gastrodin-induced ROS-mediated cell death and apoptosis in GBM cells. Moreover, a significant increase in executioner caspase 3/7 activity in Olt-treated cells as compared to Ctrl suggested that the Olt-induced apoptosis was executed through caspase 3/7 activation, which is in line with similar studies conducted using gingerol, silymarin, and CHBC—an indole derivative [20,39]

The GBM malignancy is also augmented by the presence of a niche population of cells called CSCs that possess an enormously high potential for tumorigenicity. The presence of CSCs has been verified in many cancers, including GBM. They possess self-renewal, multipotent differentiation properties, and the capacity to generate new tumors [4]. CSCs have been attributed to aggressiveness, relapse, and resistance to chemotherapy in GBM [40]. CD44 is widely used as a marker for the identification of CSCs in various cancers, including that of the brain [4,41,42,43,44,45]. In fact, its expression level is negatively correlated with GBM patients’ survival [45]. Similarly, Aldehyde dehydrogenases (ALDH), a group of enzymes that are involved in detoxifying aldehydic products produced by ROS and that, hence, contribute to cell survival, have been ascribed to the CSC phenotype [7,46,47,48]. Moreover, ALDH enzyme activity is important for eliciting its response in chemo-resistance, cell proliferation, and differentiation. In fact, several studies have indicated a decrease in ALDH^+^ cells in response to various curative and preventive treatments [47,48]. In the human prostate cancer cell line, LNcaP, PC-3luc, and the inhibition of ALDH1 were reported to induce differentiation in vitro and impair clonogenicity [47]. Our findings with the Olt exposure of GBM showing a significant reduction in the expression of CD44, ALDH, and other stem cell markers, such as Oct4 and Nanog, reflected a decrease in the stemness of CSCs and a directive towards differentiation. Moreover, the repressor effect of Olt exposure on both CD44^+^ and CD44^−^ populations suggested that the Olt was effective in containing GSCs as well as non-GSCs. 

One way to contain cancer would be to target the CSCs and induce them to differentiate. Lately, many studies have focused on targeting CSCs to remove cancer cells by regulating stem cells with a therapeutic strategy called differentiation therapy [49,50,51]. Hence, a significant reduction in the CSC population marked by a decrease in the expression of stem cell markers, reduced sphere-forming ability characterized by a decline in the number of spheres as well as a reduction of sphere area also validated the antagonistic effect of Olt on CSCs. Similar results have been reported in many other studies where chemotherapeutic drugs and natural compounds have caused differentiation of CSCs by inducing apoptosis [5,42,43,48]. The natural compound curcumin was also found to decrease the malignant characteristics of GSCs and glioma-initiating cells via the induction of ROS and autophagy [52,53]. Nestin is one of the intermediate filament proteins detected abundantly in neuroepithelial stem/progenitor cells in the growing central nervous system [23]. Hence, the loss of Nestin expression and enhanced GFAP expression accompanied by increased GFAP^+^ cells seen following Olt treatment in our experiment was indicative of cellular differentiation. 

In the case of solid tumors such as GBM, the process of EMT is commonly linked with aggressiveness, relapse, and the metastasis of cancer. Moreover, its induction in cancer causes the acquisition of the CSC-like phenotype. Having detected the efficacy of Olt in vitro and in vivo, we further wanted to study the effect of Olt on EMT markers β-catenin and vimentin. β-catenin is known to interact with the members of the LEF/TCF family of transcription factors and mediates the trans-activation of genes involved in metastasis and invasion. A study has shown the presence of TCF/LEF-1-binding motifs in the Vimentin promoter [54]. Hence, canonical Wnt activation mediated β-catenin nuclear translocation and binding to these motifs might activate Vimentin expression and, therefore, EMT induction [55]. This may support the role of β-catenin and Vimentin in EMT and malignant transformation. Accordingly, the Olt-mediated reduction in β-catenin and Vimentin expression seen in our study might facilitate reversing the process of EMT.

In many recent articles, it has been reported that CD44 is a downstream target of Wnt signalling and is widely proven to maintain the stemness of GSCs [22,56]. In gastric cancer, CD44 was shown to modulate Wnt/β-catenin signaling, which is primarily involved in tumor metastasis and progression [57]. The authors have suggested that CD44 downregulation reduces its interaction with N-WASP and ErbB2. This eventually affects the phosphorylation of β-catenin and inhibits actin polymerization to reduce cancer cell migration. Hence, downregulation of CD44 in cancer cells would inhibit migration and the invasion of cells through the impairment of β-catenin expression. Thus, we speculated that a decrease in the expression of CD44 in GBM cells would reduce migration and the invasion of cancer cells by modulating β-catenin. Further investigation of the modulation of Wnt signaling by Olt would address this.

In line with our in vitro findings, the ectopic model for GBM using SCID mice in vivo also validated the efficacy of Olt in containing GBM. The Olt-administered group showed a significant reduction in tumor growth within the monitored time regimen and without rendering any systemic toxicity in any of the vital organs tested. This was ascertained by monitoring no significant changes in biochemical and hematological parameters as well as in tissue architecture, hence suggesting its suitability for patient usage without any adverse effects on other organs. These findings corroborated well with our in vitro cell viability assay performed with HEK293 cells and MSCs, where the Olt did not adversely impact cell growth and viability. Similar results have also been reported by several other natural compounds such as curcumin, epoxyazadiradione, Magnolol, Kukoamine A, AECHL-1, etc. [7,26,52,58,59]. Collectively, our data suggested the potential anti-tumorigenic propensity of Olt that can serve as a plausible adjuvant therapeutic for containing GBM.

## 5. Conclusions

The lingering bottleneck in oncotherapy pertains to chemo-resistance and toxicity to normal cells. Although a majority of chemotherapeutics are targeted to induce apoptosis within a tumor, non-specific effects on normal cells as well as bi-allelic inactivation in cancer cells pose a major hindrance to the same. Hence, strategies to curb cancer cell proliferation induce autophagy in parallel to differentiation induction may serve as plausible alternatives. Our investigation using the Olt demonstrated the potential of Olt to induce apoptosis in GBM cells by modulating tubulin arrangements and arresting the cell cycle, whereas it remained non-toxic to normal cells. In fact, its anti-cancerous effect in vitro was either akin to or superior to the standard drug Cispln. Undoubtedly, the encouraging results obtained with the Olt from both in vitro and in vivo studies have formed a strong basis for its further exploration and usage as an advanced anticancer adjuvant drug candidate for the possible prevention and treatment of GBM and other cancers using patient samples and patient-derived xenograft models. Notwithstanding, an advanced study on the mechanistic action of the Olt and its clinical efficacy may render its bedside translation.

## Figures and Tables

**Figure 1 cells-11-03057-f001:**
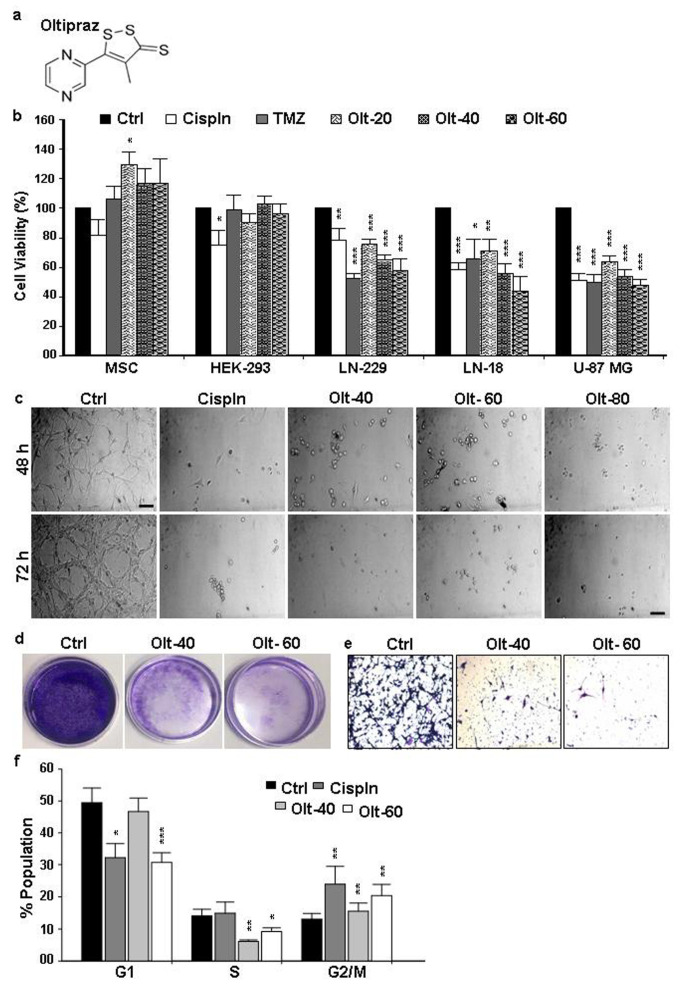
The inhibitory effect of Olt on GBM cell growth: (**a**) The chemical structure of Olt. (**b**) MTT assay depicting the % cell viability in normal (MSC), non-tumorigenic (HEK-293), and GBM (LN18, LN229, and U-87 MG) cell lines after 48 h of Olt treatment (20–60 μM) compared to the untreated control (Ctrl) group in each (considered as 100% viable). The TMZ and Cispln exposure served as the positive controls for treatment groups in GBM cells. Values are represented as mean ± SEM from independent experiments (n = 3–7). (**c**) Morphological changes in U-87 MG cells after treatment with Cispln and Olt (40-, 60- and 80 μM) for 48- and 72 h (Scale bar 50 µM). (**d**) Decrease in colony formation abilities of GBM cells after treatment of the Olt for 48 h. (**e**) Olt treatment (48 h) inhibited the migration of cells from the upper chamber to the lower chamber of the transwell plate. Scale bar: 100 μM. (**f**) Flow cytometry analysis shows Olt-induced reduction in G1 and S phases and G2/M phase arrest in the cell cycle in U-87 MG cells. Data are presented as mean ± SEM (n = 6). * *p* ≤ 0.05; ** *p* ≤ 0.01; *** *p* ≤ 0.001 compared to the untreated Ctrl group.

**Figure 2 cells-11-03057-f002:**
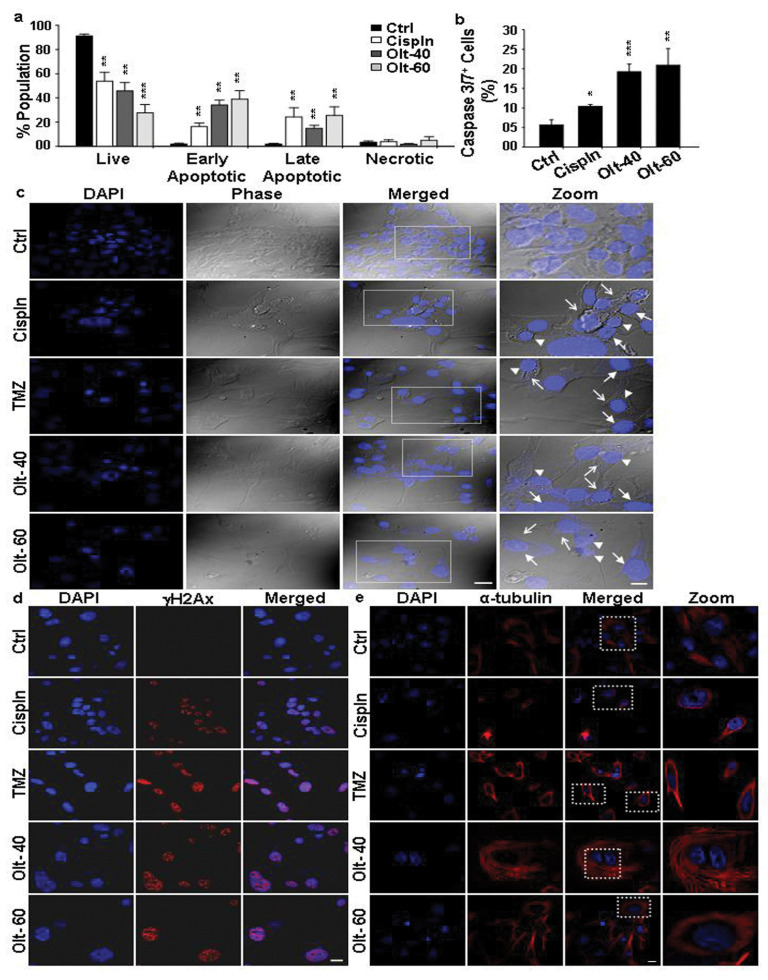
Pro-apoptotic influence of Olt on U-87 MG cells: (**a**,**b**) Flow cytometry quantification of (**a**) Annexin V/APC–PI and (**b**) Caspase 3/7 stained population indicates an increase in both early (Annexin V^+^/PI^−^) and late (Annexin V^+^/PI^+^} apoptotic cells and Caspase 3/7^+^ cells following Olt exposure of U-87 MG cells for 48 h. Cells treated with Cispln served as the positive control for the treatment group. Data are presented as mean ± SEM (n = 3–7). * *p* ≤ 0.05; ** *p* ≤ 0.01; *** *p* ≤ 0.001 compared to the untreated Ctrl. (**c**) Hoechst 33342 staining reveals chromatin condensation (arrowhead), nuclear fragmentation (closed arrow), and membrane blebbing (open arrow) seen in U-87 MG cells treated with TMZ, Cispln (both positive controls), and Olt (40-, and 60 μM) for 48 h. Magnified views (Zoom) of the boxed regions are shown with contrast adjustment for better visualization. (**d**) γH2αX staining of U-87 MG cells displaying punctate nuclei in the treated groups reflects nuclear fragmentation upon exposure of Cispln, TMZ, and the Olt for 48 h. (**e**) Microtubules’ rearrangement was detected by α-tubulin staining in 48 h post-treated (Cispln, TMZ, Olt) U-87 MG cells. Magnified views (Zoom) of the boxed regions are shown for better visualization. Scale bar: 20 μM (**c**–**e**), 10 μM (zoomed images in **c**,**e**).

**Figure 3 cells-11-03057-f003:**
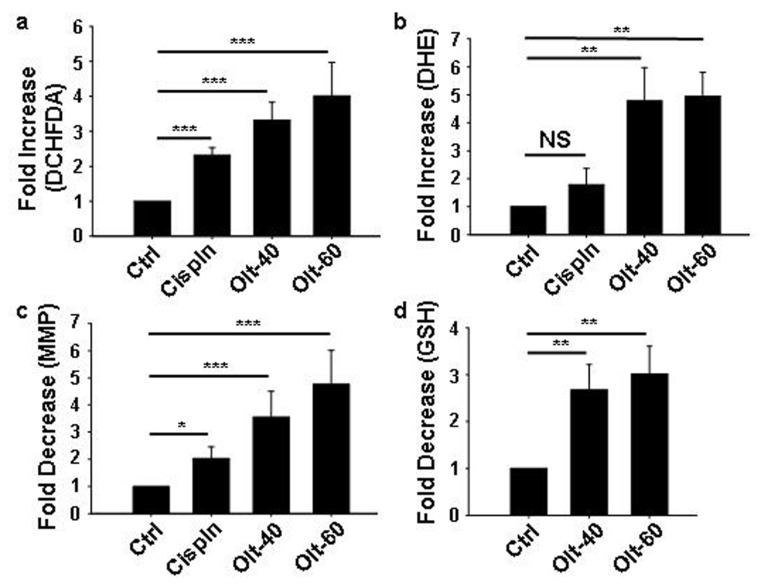
Influence of Olt on ROS and mitochondrial membrane potential: (**a**,**b**) Flow cytometry quantification of DCHFDA^+^ and DHE^+^ U-87 MG cells representing ROS-high state, following their exposure to Cispln and two different concentrations of Olt (40-, and 60 μM). The data shows the fold increase in ROS in Cispln- and Olt-treated cells compared to untreated Ctrl. (**c**) Flow cytometry detection of loss of MMP by Rhodamine-123 staining. The data shown represent the fold difference in MMP loss in Cispln- and Olt-treated cells compared to untreated Ctrl. (**d**) Flow cytometry quantification of GSH reveals an Olt-mediated decrease in glutathione levels following Olt exposure of U-87 MG cells for 48 h. Data shown represents the fold decrease in glutathione levels in the Olt-treated cells compared to untreated Ctrl. Data are presented as mean ± SEM (n = 4–8). * *p* ≤ 0.05; ** *p* ≤ 0.01; *** *p* ≤ 0.001.

**Figure 4 cells-11-03057-f004:**
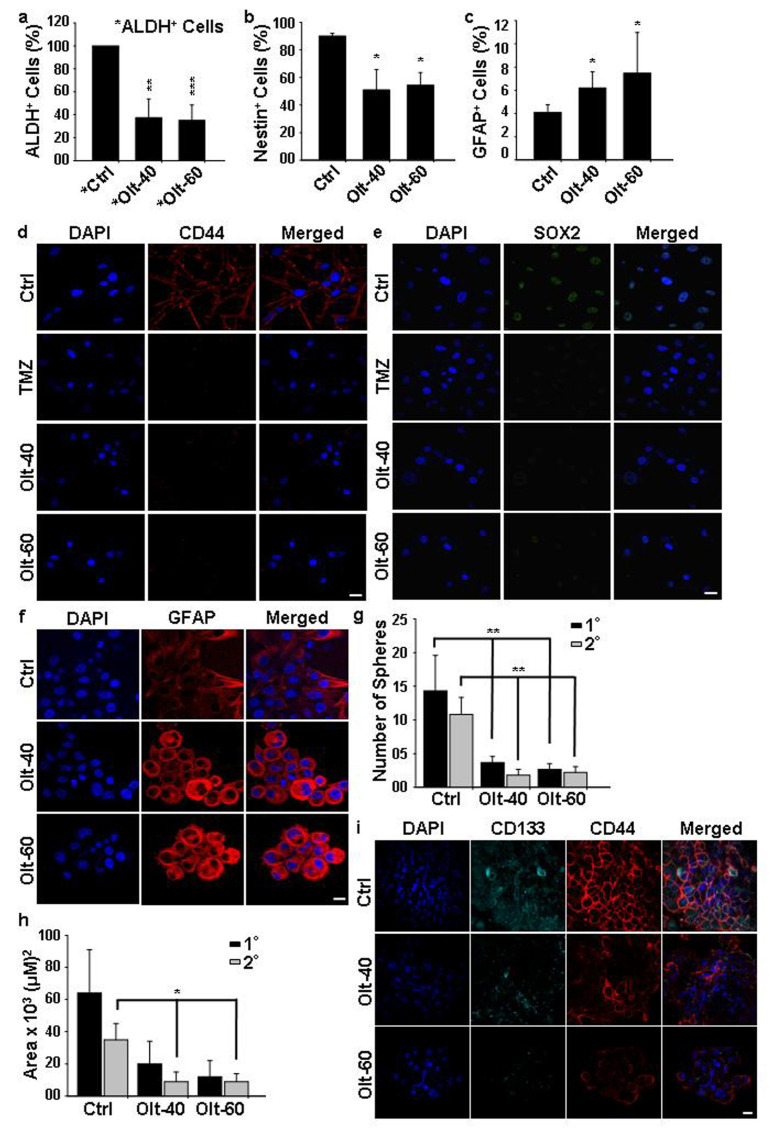
Influence of Olt on GSC population in vitro: (**a**) The ALDH^+^ cells representing the GSCs pool from U-87 MG cells, following their purification by FACS and reseeding, displayed a significant reduction in ALDH^+^ cells upon their exposure to the Olt for 48 h. (**b**,**c**) Olt exposure of U-87 MG cells for 48 h in culture decreased the number of Nestin^+^ cells (**b**) but increased the GFAP^+^ cells **(c**). (**d**,**e**) The immunostained pattern also revealed a reduction in U-87 MG cells exhibiting stem cell markers such as CD44 (**d**) and SOX2 (**e**) expression upon Olt exposure. (**f**) Olt exposure of U-87 MG cells led to an increase in GFAP, a mature astroglial marker, expression in them compared to untreated Ctrl. (**g**,**h**) Decrease in the number (**g**) and size (**h**) of primary (1°) and secondary (2°) oncospheres in U-87 MG cells post-treatment of the Olt for 48 h. (**i**) Immunostaining of CD44 and CD133 in oncospheres developed showed a decrease in the expression of the CD133 and CD44 populations in the Olt-treated U-87 MG cells. Scale bar: 20 µM. Data are presented as mean ± SEM (n = 4 in **a**–**c**, and 3 in **g**,**h**). * *p* ≤ 0.05, ** *p* ≤ 0.01, *** *p* ≤ 0.001 compared to the untreated Ctrl.

**Figure 5 cells-11-03057-f005:**
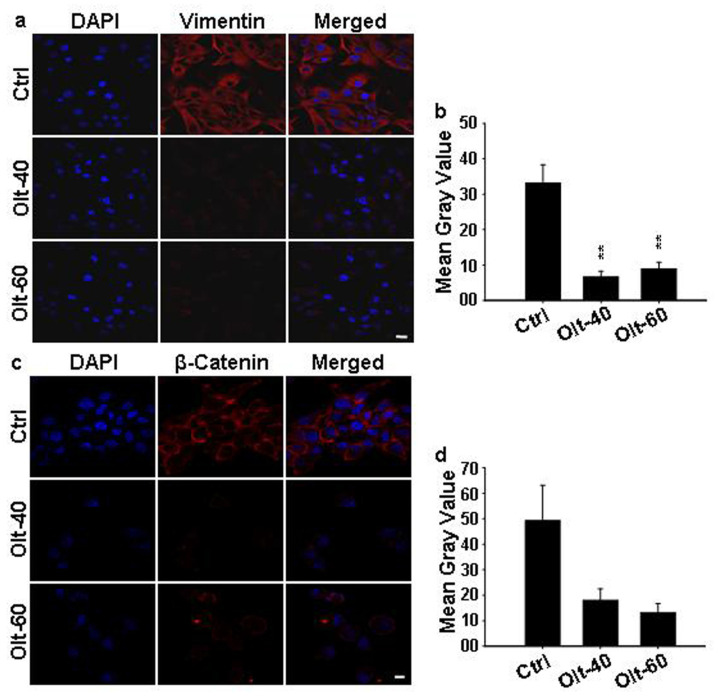
Influence of Olt on EMT reversal**:** (**a**,**c**) Immunofluorescence examination of EMT marker vimentin (**a**) and β-catenin (**c)** expression in U-87 MG cells revealed a reduction in the same following their exposure to Olt. Scale bar: 20 μM. (**b**,**d**) Respective intensity quantification from (**a**) and (**c**) (from 3 different fields). Data are presented as mean ± SEM (n = 3–4). ** *p* ≤ 0.01.

**Figure 6 cells-11-03057-f006:**
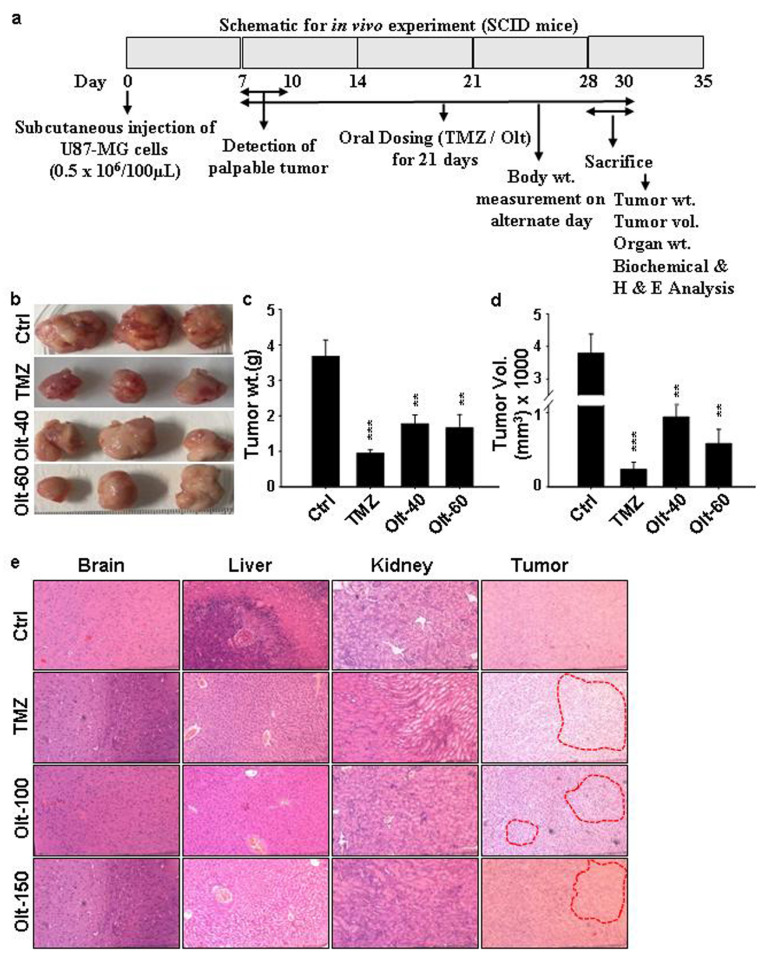
Anti-tumorigenic efficacy of Olt in vivo. TMZ (positive control) and Olt administration for 3 w, at the stated doses, led to reduced tumor progression in the U-87 MG cell-induced GBM xenograft mouse model. (**a**) The schematic for in vivo experiment. (**b**) Representative images of tumors excised from different groups (n = 5/group). (**c,d**) Measurement of tumor weight (**c**) and volume (**d**) in each group post-sacrifice. Data are presented as mean ± SEM ** *p* ≤ 0.01, *** *p* ≤ 0.001 compared to the disease control (Ctrl) group. (**e**) Representative images of H&E-stained tissue sections taken from tumor and stated vital organs from disease control (Ctrl) and U-87 MG cell-induced GBM tumor-bearing mice.

**Table 1 cells-11-03057-t001:** Body Weight of tumor-bearing mice after oral administration of Olt for 21 days.

Groups	Initial/Day 0	Week-I	Week-II	Week-III
Ctrl	21.5 ± 1.73	21.9 ± 1.21	21.3 ± 1.24	22.5 ± 1.58
TMZ-50 *	21.5 ± 1.96	21.9 ± 1.36	20.5 ± 1.41	21.8 ± 1.09
Olt-100 *	21.3 ± 2.31	20.0 ± 1.02	20.3 ± 0.74	20.5 ± 0.86
Olt-150 *	20.4 ± 1.56	20.8 ± 1.07	20.9 ± 0.82	21.1 ± 1.40

* Drugs in mg.

**Table 2 cells-11-03057-t002:** Relative organ weight (g%) of tumor-bearing mice administered with the Olt for 21 days post-sacrifice.

Organs	Ctrl	TMZ-50 *	Olt-100 *	Olt-150 *
Brain	1.52 ± 0.30	1.89 ± 0.11	2.16 ± 0.14	2.06 ± 0.22
Liver	4.27 ± 0.93	5.78 ± 0.54	5.38 ± 0.40	5.86 ± 1.54
Kidney	1.15 ± 0.27	1.39 ± 0.13	1.35 ± 0.07	1.34 ± 0.20
Heart	0.44 ± 0.10	0.68 ± 0.09	0.57 ± 0.09	0.52 ± 0.07
Lungs	0.93 ± 0.24	0.95 ± 0.16	1.28 ± 0.19	1.17 ± 0.19
Spleen	0.80 ± 0.24	0.45 ± 0.03 *	0.76 ± 0.12	0.56 ± 0.21

* Drugs in mg.

**Table 3 cells-11-03057-t003:** Assessment of serum biochemical parameters in tumor-bearing mice administered with the Olt for 21 days post-sacrifice.

Biochemical Parameters	Ctrl	TMZ-50 *	Olt-100 *	Olt-150 *
Total Bilirubin (mg/dL)	0.42 ± 0.04	0.47 ± 0.05	0.60 ± 0.20	0.57 ± 0.08
Bilirubin Direct (mg/dL)	0.25 ± 0.03	0.28 ± 0.04	0.30 ± 0.05	0.26 ± 0.03
Bilirubin Indirect (mg/dL)	0.17 ± 0.02	0.15 ± 0.02	0.30 ± 0.10	0.23 ± 0.03
SGPT (IU/L)	34.67 ± 17.33	43.08 ± 5.84	55.63 ± 6.71	44.66 ± 8.17
SGOT (IU/L)	108.32 ± 15.2	122.28 ± 22.91	93.00 ± 21.0	131 ± 77.7
Blood Urea (mg/dL)	39.77 ± 6.14	44.87 ± 3.48	38.33 ± 4.40	38.5 ± 3.06
Blood Urea nitrogen (mg/dL)	18.58 ± 2.87	20.96 ± 1.62	17.91 ± 2.06	17.99 ± 1.43
Uric Acid (mg/dL)	1.3 ± 0.33	1.75 ± 0.09	1.60 ± 0.20	1.60 ± 0.34

* Drugs in mg.

## Data Availability

Not Applicable.

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
