# Peer review of "Elucidating the Anti-Tumorigenic Efficacy of Oltipraz, a Dithiolethione, in Glioblastoma"

_cells, 2022, doi:10.3390/cells11193057_

Round 1

Reviewer 1 Report

The manuscript by Kapoor-Narula and Lenka describes the anti-tumorigenic properties of a phytochemical compound, namely the dithiolethione Oltipraz (Olt). While the anticancer effects of Olt have been already established and observed in various tumor entities, little or no data have been published reporting its efficiency in suppressing brain tumors. Therefore, the present manuscript brings certain novelty to the field of cancer (in particular glioblastoma) therapy research. The Abstract summarizes well the authors’ findings, including the outcome of the in vitro and in vivo experiments. The mechanisms of Olt antitumor action are also elucidated, which may be important if Olt -based combinatory treatments are to be considered.

Introduction:

The Introduction is well written and comprehensive. (1) However, it would be useful if the authors would shortly explain how the balance between Olt antioxidant and pro-oxidative effects works and how activation of an antioxidant response in normal cells complies with oxidative stress (e.g. ROS) induction in cancer cells. Minor: (2) Does the sentence before references 2 and 3 in the first paragraph contain a typing mistake? (3) Overexpression is usually written as single word.

Materials and Methods:

(1) In the Materials and Methods section, the source and the catalog number are indicated for the U87-MG cell line only, but not for LN-229 and LN-18. (2) What was the reason for having selected HEK-293 cells as controls? This is still a virus-transformed cell line, i.e. it is hardly the best representative of the normal, non-tumor cell population in vivo. Could not a primary glial cell culture be used instead? It may be that the authors had a specific reason for this choice; this has to be then clearly justified in the text. (3) In 2.4.5. (a) -  Annexin-PI, and (b) –Caspase 3/7, are not indicated. (4) In 2.6.2 (CD44 estimation): Why did not the authors use patient-derived GBM cells? That would be much more convincing than the use of an immortalized culture, such as U87-MG. Why was the choice made for CD44 and ALDH, and another typical marker, Olig2, was not included? (5) Another question in regard to this section concerns the usage of the three GBM cell lines, as mentioned in the Abstract. All three cell lines were used only for the assessment of cellular viability. In all further experiments, only the U87-MG cell line was used, although it would be important to confirm the observations made in U87-MG cells also in the other two GBM lines. (6) The animal experiments are carefully made, with all controls required and studying a wide variety of parameters in the treated mice. However, the ectopic subcutaneous model offers only some indicative observations as it parallels distantly to the real in vivo GBM situation. The lack of functional immune system in SCID mice and the exclusion of the blood-brain-barrier as an important factor in determining GBM specificities makes the subcutaneous model useful, but not decisive in evaluating the efficiency of a certain drug against this tumor.

Results:

(1) By determining the effects of Olt on cell growth characteristics it would be useful to combine the MTT test with another cell viability assay demonstrating for example the preservation (or not) of cell membrane integrity, cellular ATP contents, etc. What was the reason of not reaching the 50% cytotoxicity dose (CCID50)? As illustrated on Fig. 1a, CCID50 is approached in U87-MG, but not reached in any of the three cell lines tested. What was the reason for the authors not to decide for a further increase of the dose if no toxicity was supposedly observed with e.g. 80 µM of Olt? Furthermore, the effects of Olt-60 in U87-MG cells are comparable to the ones of Cisplatin and TMZ. (2) Coming back to the normal control issue mentioned above: TMZ is similarly non-toxic for HEK cells, i.e. in this particular case it is not an appropriate control. (3) On Fig. 1c it would be useful not to omit cisplatin as a positive control, but show its inhibitory effects on GBM colony formation abilities. (4) On Fig. 2a, it would be interesting to see the comparison or early-late apoptotic and necrotic populations in cicplatin-treated U-87MG cells. Rescue experiments with various apoptosis inhibitors could be also interesting. (5) In the GSC paragpragh, the authors should maybe avoid the usage of the “normal cancer cells” expression… (6) In the in vivo experimental setting, what was the reason for choosing TMZ and not Cisplatin as a control?

Discussion:

May be slightly shortened

Author Response

Please see the attachment (Response to Reviewer-1).

Reviewer 2 Report

The paper need some modification - 

1) World-wide, an estimated 251,329 people have died from primary cancerous brain and CNS tumors in 2020 [1]. "Author should provide some data from his own country to establish that the research done is with purpose not as me to work aim. 

2) Author should discuss some previous therapies for Glioblastoma and their outcome and limitation. 

3) provide the structure of Olt using chem draw

4) Animal ethical number should be provided. 

5) Author has written Material and Methods but where are the details of the materials used?

6) The writing style is very vague and has many grammatical errors. 

7) provide graphical abstract 

8) working scheme for the experiment including animal study

9) why 70 references in research articles, reduce it to 50-60 

Author Response

Please see the attachment (Response to Reviewer-2).

Reviewer 3 Report

The manuscript entitled Elucidating the anti-tumorigenic efficacy of Oltipraz, a dithiolethione, in Glioblastoma addresses a topical issue in glioblastoma research area, yet it needs some improvements in order to be published.

Thus, the authors should added:

-       at the Chapter 2.1 Cell culture and treatments, they have to mention all the cell lines, including LN18, LN229, HEK-293, not only U87. Also the Olt origine must be mentioned in this part.

-       at the Chapter 2.2. should be mention the cell viability for long term exposure, not only 24 and 48 hours

-       the P-value in Chapter 2.10 must be in accordance with the text from figure 1 were we have p-value

-       Future research directions should be highlighted.

Considering the above mentioned suggestions, we recommend the publication of the manuscript after revisions are made; still the final decision belongs to Editor-in-chief.

Author Response

Please see the attachment (Response to Reviewer-3).

Round 2

Reviewer 1 Report

The revised version of the manuscript complies sufficiently with the suggestions made and gives a reasonable justification for some methodological weaknesses.

Reviewer 2 Report

The corrections are addressed